# Genome-Wide Identification of Peanut *B-Boxs* and Functional Characterization of *AhBBX6* in Salt and Drought Stresses

**DOI:** 10.3390/plants13070955

**Published:** 2024-03-26

**Authors:** Haohong Tang, Cuiling Yuan, Haonan Shi, Feng Liu, Shihua Shan, Zhijun Wang, Quanxi Sun, Jie Sun

**Affiliations:** 1Key Laboratory of Oasis Eco-Agriculture, College of Agriculture, Shihezi University, Shihezi 832000, China; tanghaohongwf@163.com (H.T.); shihaonanxj@163.com (H.S.); liufeng@shzu.edu.cn (F.L.); 2Shandong Peanut Research Institute, Qingdao 266100, China; yuancl1982@163.com (C.Y.); shansh1971@163.com (S.S.); 3Biotechnology Research Institute, Xinjiang Academy of Agricultural and Reclamation Science, Shihezi 832000, China; wzjshihezi@163.com

**Keywords:** peanut, *BBX* gene family, abiotic stress, VIGS

## Abstract

The *B-box* (*BBX*) gene family includes zinc finger protein transcription factors that regulate a multitude of physiological and developmental processes in plants. While *BBX* gene families have been previously determined in various plants, the members and roles of peanut *BBXs* are largely unknown. In this research, on the basis of the genome-wide identification of *BBXs* in three peanut species (*Arachis hypogaea*, *A. duranensis*, and *A. ipaensis*), we investigated the expression profile of the *BBXs* in various tissues and in response to salt and drought stresses and selected *AhBBX6* for functional characterization. We identified a total of 77 *BBXs* in peanuts, which could be grouped into five subfamilies, with the genes from the same branch of the same subgroup having comparable exon–intron structures. In addition, a significant number of *cis*-regulatory elements involved in the regulation of responses to light and hormones and abiotic stresses were found in the promoter region of peanut *BBXs*. Based on the analysis of transcriptome data and qRT-PCR, we identified *AhBBX6*, *AhBBX11*, *AhBBX13*, and *AhBBX38* as potential genes associated with tolerance to salt and drought. Silencing *AhBBX6* using virus-induced gene silencing compromised the tolerance of peanut plants to salt and drought stresses. The results of this study provide knowledge on peanut *BBXs* and establish a foundation for future research into their functional roles in peanut development and stress response.

## 1. Introduction

Peanuts (*Arachis hypogaea* L.), rich in protein and oil, hold significance as an important oilseed and valuable economic crop worldwide. With the rising global population, the demand for peanut oil and other products derived from peanuts is gradually increasing. Global warming has exacerbated drought and soil salinization, which are major constraints for increasing peanut yield and quality. As such, understanding the genetic and molecular pathways underpinning peanut responses to abiotic stresses has become a primary objective in addressing these issues when faced with the production of not only peanuts but also other crops worldwide.

The *B-box* (*BBX*) family is a subfamily of the zinc-finger structural protein family. B-box proteins can bind to RNA and DNA through their interaction with zinc ions to regulate gene expression. The *B-box* gene family has been extensively studied in numerous plant species [1,2,3,4,5,6,7,8]. In *Arabidopsis*, CONSTANS (CO) was identified as the first plant *BBX* involved in photoperiod-dependent flowering [9]. Subsequently, more *BBX* members such as *AtBBX4* [10], *AtBBX5* [11], *AtBBX6* [12], *AtBBX7* [13], *AtBBX28* [14], and *AtBBX32* [15] were also found to play important roles in controlling plant flowering.

*BBXs* regulate the biological processes of not only flowering but also responding to abiotic stresses [14]. The salt tolerance protein STO, encoded by *AtBBX24*, was discovered to have a positive impact on the salt stress response, and STO-OE improves salt tolerance in *Arabidopsis* [16]. In chrysanthemum (*Chrysanthemum morifolium*), the suppressed expression of *CmBBX24* is linked to a decrease in the plant’s ability to withstand cold and drought conditions [17]. The expression of *CmBBX19* is down-regulated in chrysanthemums under drought stress and ABA treatment [18]. Overexpressing *CmBBX19* results in reduced drought tolerance [18]. In rice, the expression of *OsBBX1* and *OsBBX24* was found to cause significant changes in response to abiotic stresses [19]. Overexpressing *MdBBX7* enhances drought tolerance in apples [20]. A different study on apples discovered that by interacting with MdABI5 (the protein that regulates ABA signals) and MdEIL1 (the protein that regulates ethylene signals), *MdBBX37* controls the process of leaf senescence mediated by ABA and ethylene [21]. In tomatos, overexpressing *SlBBX17* enhances cold tolerance, while silencing *SlBBX17* increases cold susceptibility [22]. Taken together, these findings indicate that *BBXs* are significant factors in controlling plant growth and development, as well as in regulating responses to abiotic stresses. Despite the findings on the role of *BBXs* in many plants [16,17,18,19,20,21,22], there is limited knowledge about the *BBXs* in peanuts and their functions in the development and stress response of peanuts.

The cultivated peanut (*Arachis hypogaea* L.) is an allotetraploid (AABB, 2n = 4x = 40) [23,24]. The availability of the peanut reference genome sequence [25] enables the genome-wide identification and analysis of gene families.

In this study, we analyzed the peanut *BBX* gene family, including phylogenetic relationships, the gene structure, and *cis*-acting elements. Additionally, we explored the expression patterns of *AhBBXs* in various tissues and in response to abiotic stress using publicly accessible RNA-Seq data. The main objectives of this research were to establish a comprehensive theoretical and technical framework that enhances our comprehension of the functional role of *AhBBXs*, particularly in drought and salt tolerance.

## 2. Results

### 2.1. Identification and Physicochemical Analysis of AhBBXs

A total of 39 (*A. hypogaea*), 19 (*A. duranensis*), and 19 (*A. ipaensis*) *BBXs* were identified from the genomes of the cultivated and the two wild peanut species. The number of *BBXs* in the cultivated peanut was roughly the sum of the numbers in the two wild peanut species. The *BBXs* were given the names *AdBBX1-AdBBX19*, *AiBBX1-AiBBX19*, and *AhBBX1-AhBBX39*, respectively, based on their chromosomal locations.

The length of peanut BBX proteins ranged from 154 amino acids (*AiBBX17*) to 546 amino acids (AdBBX14). The BBX proteins had a molecular weight range from 16.98 kDa (AiBBX17, AhBBX37) to 61.01 kDa (AdBBX14). The predicted isoelectric points (*pI*) ranged from 4.12 (AiBBX17, AhBBX37) to 9.41 (AhBBX28), with most of the members having an isoelectric point less than seven. Only nine members, AdBBX7 (*pI* 7.12), AdBBX15 (*pI* 8.19), AiBBX7 (*pI* 8.59), AiBBX11 (*pI* 7.47), AiBBX15 (*pI* 8.88), AhBBX12 (*pI* 8.19), AhBBX17 (*pI* 8.86), AhBBX28 (*pI* 9.41), and AhBBX35 (*pI* 9.13), had isoelectric points above seven. Except for *AdBBX14*, which was predicted to be placed on the cell membrane, all other BBXs were predicted to be located in the nucleus (Appendix A).

### 2.2. Phylogenetic Analysis of Peanut BBXs

To obtain a deeper understanding of the phylogenetic relationships of peanut BBX proteins, the 77 BBX protein sequences from wild and cultivated peanuts were compared with those from *Arabidopsis* and rice. A phylogenetic tree was generated using sequence alignment (Figure 1, Appendix A).

The B-box and CCT domains were used to divide the 77 peanut BBX proteins into five major subfamilies (Appendix A). Subfamily I (Group I) BBXs consisted of two B-box structural domains and one CCT structural domain, including four AdBBXs, four AiBBXs, and eight AhBBXs. Subfamily II (Group II) BBXs also contained two B-box structural domains and one CCT structural domain, including one AdBBX, one AiBBX, and four AhBBXs, but the amino acid sequence of the B-box 2 domain was somewhat different between the two groups. Subfamily III (Group III) BBXs contained one B-box structural domain and one CCT structural domain, including three AdBBXs, three AiBBXs, and six AhBBXs. Subfamily IV (Group IV) BBXs contained only two B-box structural domains, including eight AdBBXs, seven AiBBXs, and 12 AhBBXs. Subfamily V (Group V) BBXs contained only one B-box structural domain, including three AdBBXs, four AiBBXs, and nine AhBBXs.

### 2.3. Chromosome Locations and Gene Duplication Prediction of Peanut BBXs

*BBXs* are distributed on all chromosomes except chromosomes A2, B2, and B7 of the two wild species and chromosomes 2, 12, and 17 of the cultivated species, with most of the genes distributed at chromosome ends (Figure 2). Among them, chromosome 13 contains the most *BBXs*, with six genes. Chromosomes A3 and B3 each have five genes, while chromosomes A10, B8, B10, 6, 16, and 20 each have three genes. Chromosome 3 has four genes, and the remaining chromosomes contain one or two *BBXs*.

Segment duplications and tandem duplications have been proposed as primary factors in gene family expansion in previous studies [26]. Hence, the *BBXs* in the cultivated peanut were analyzed for segmental duplications or genome-wide duplication events. The 39 *AhBBXs* generated a total of 43 sets of segmental duplications (Figure 3, Appendix A), suggesting that the emergence of certain peanut *BBXs* might be attributed to duplication events.

### 2.4. Syntenic Analysis

To investigate the phylogenetic relationship between *AhBBXs*, we constructed collinearity maps between peanuts and four typical species as follows: *Arabidopsis***,** soybeans, tomato, and rice (Figure 4). In this study, a total of 31 *AhBBXs* showed collinearity with tomato *BBX* genes, followed by 29 in soybeans, 24 in *Arabidopsis* and 11 in rice (Appendix A). Among the identified orthologos, 101 pairs belonged to soybeans, 48 pairs to tomato, 32 pairs to *Arabidopsis*, and 17 pairs to rice (Appendix A). In addition, seven genes had gene pairs in all four species, suggesting that they might have existed before species differentiation. Interestingly, some genes identified as collinear pairs between peanuts and three dicotyledonous plants were not found in rice, suggesting that these collinear pairs may have appeared after the differentiation of dicotyledonous and monocotyledonous plants.

### 2.5. Gene Structure and Conserved Motif Analysis of Peanut BBXs

A maximum likelihood phylogenetic analysis was performed utilizing peanut BBX proteins (Figure 5A) to link different groups to gene structure, conserved protein motifs (Figure 5B, Appendix A) and exon–intron structures (Figure 5C). It was found that the number of exons in BBXs varied from one (*AdBBX15*) to six (*AhBBX28*). Among them, six genes contained one exon, 35 genes contained two exons, 20 genes contained three exons, nine genes contained four exons, six genes contained five exons, and one gene contained six exons.

According to the phylogenetic tree and gene structure analysis, genes located in the same branch usually have similar gene structures. For example, *AiBBX18*, *AdBBX18*, *AhBBX19*, and *AhBBX38* all contain two exons with the same exon position. Similarly, *AiBBX10*, *AdBBX10*, *AhBBX10*, and *AhBBX30* all have three exons in similar positions. Furthermore, the genes of the same subfamily in the evolutionary tree exhibit similar gene structures, implying potential functional resemblances among *BBXs* from the same subfamily.

The examination of the conserved structural domains indicated that the *BBX* family members possess between three and seven conserved structural motifs. All members have the core sequence of the B-box 1 domain (motif 1), except *AhBBX28*. In total, 49 members contain the B-box 2-conserved domain (motif 3, motif 4), and 32 members contain the CCT domain (motif 2) in addition to the typical B-box domain. It remains to be investigated whether the deletion of these motifs confers some function specific to the protein, though the highly conserved motifs among peanut BBX proteins prove useful for an analysis of their functionality.

### 2.6. Analysis of cis-Acting Elements in the Promoters of the Peanut BBXs

The expression of genes is mainly regulated by promoters. Gene responses to external environmental or internal developmental signals can be predicted using *cis*-acting elements in the promoter region. Therefore, we conducted predictive analysis for *cis*-acting elements using the 2000 bp sequence upstream of the translation start site of the *BBXs* and identified 1639 such elements (Figure 6, Appendix A).

Almost all peanut *BBX* promoter regions were found to be rich in *cis*-acting elements related to responses to hormones such as ABA, MeJA, gibberellins (GA), and salicylic acid (SA), as well as responses to adverse stresses like drought and low temperature. Among them, at least 51 elements were present in the *AdBBX15* promoter sequence. The *AiBBX10* promoter contains at least 61 elements. The *AhBBX30* promoter contains at least 65 elements, comprising 19 components sensitive to ABA and 32 components responding to light, as well as a number of other elements responsive to MeJA and others. The most abundant *cis*-acting elements found in the promoter sequences of peanut *BBXs* are the ones that react to light, MeJA, and ABA, accounting for 42% (687), 13% (220), and 13% (219), respectively. The implication is that peanut *BBXs* might have significant functions in response to light signaling pathways and adverse stress [27].

### 2.7. Expression of Peanut BBXs in Different Tissues

By utilizing publicly available RNA-seq data, we studied the expression levels of peanut *BBXs* in 22 tissues from various stages of development. (Figure 7, Appendix A). After processing the log2-transformed FPKM results, we discovered that peanut *BBXs* were expressed in 22 distinct tissues with varying expression levels. This disparity suggests potential differences in the functions of the individual genes. Furthermore, genes from the same phylogenetic branch demonstrated similar expression patterns.

By analyzing the expression patterns of the genes in various tissues, we generated further predictions regarding the role of peanut *BBXs*. For example, *AdBBX7*, *AiBBX7*, *AdBBX8*, and *AiBBX8* exhibited higher expression in flowers, indicating their potential involvement in flower development. *AdBBX2* and *AiBBX3* were highly expressed in vegetative shoot tips, suggesting that these genes may play a role in regulating the growth and development of the stems. *AdBBX5*, *AdBBX16*, *AiBBX16*, *AhBBX4*, *AhBBX18*, and *AhBBX36* showed high expression in nodal roots, suggesting that these genes may have a role in the control of nitrogenase synthesis and activity.

Moreover, homologous genes exhibited similar expression patterns. For example, *AdBBX10*, *AiBBX10*, *AhBBX10*, and *AhBBX30* were predominantly expressed in pods, implying that these genes may influence the development of peanut seeds. The expression levels of *AdBBX6*, *AiBBX6*, *AdBBX9*, *AiBBX9*, *AhBBX5*, and *AhBBX26* in leaves were higher than in other tissues. However, the expression of *AdBBX12*, *AdBBX19*, and *AiBBX19* in the subterranean gynophore tip was reversed. Peanut evolution involved polyploidization and gene duplication events, which might result in different expression patterns for homoeologous genes. These analytical results can provide a foundation for investigating the regulatory functions of *BBXs*.

### 2.8. Expression Patterns of Peanut BBXs under Abiotic Stresses

To comprehend how *AhBBXs* react to abiotic stresses, we analyzed the expression of peanut *BBXs* utilizing transcriptomic data during drought or high salinity stress that were previously generated by others [28,29] (Figure 8, Appendix A).

The findings revealed that, among the 39 *AhBBXs* of cultivated peanuts, 15 genes demonstrated a two-fold increase in expression under drought stress, while nine genes exhibited a two-fold increase in expression when exposed to salt stress. *AhBBX6*, *AhBBX9*, *AhBBX11*, *AhBBX13*, *AhBBX14*, *AhBBX16*, *AhBBX17*, *AhBBX27*, *AhBBX31*, *AhBBX32*, *AhBBX34*, *AhBBX35*, and *AhBBX38* showed an increase in expression under both drought and salt stress. Most genes belonging to the same subgroup showed similar expression patterns when subjected to salt or drought stress. For example, the expressions of *AhBBX7*, *AhBBX15*, *AhBBX24*, *AhBBX28*, and *AhBBX33* increased under drought stress and decreased under salt stress. The expressions of *AhBBX4*, *AhBBX25*, and *AhBBX37* decreased under drought stress and increased under salt stress. The expressions of *AhBBX6*, *AhBBX7*, *AhBBX15*, *AhBBX27*, *AhBBX28*, and *AhBBX33* were elevated more than 15-fold under drought stress, and the expressions of *AhBBX4*, *AhBBX6*, *AhBBX25*, and *AhBBX37* were elevated more than eight-fold under salt stress. These results imply that *BBXs* are probably involved in controlling a variety of plant signal transduction pathways in response to stress regulation.

### 2.9. qRT-PCR of BBXs under Salt and Drought Stresses

Earlier research has shown the significant involvement of *BBXs* in responses to hormones and different biotic or abiotic pressures [30]. To further analyze the potential role of *AhBBXs* in response to abiotic stresses, eight *BBXs* that were highly expressed based on transcriptome data under salt and/or drought stress were selected for qRT-PCR validation (Figure 9).

Under drought treatment, the expression levels of *AhBBX6*, *AhBBX7*, *AhBBX11*, *AhBBX13*, *AhBBX15*, *AhBBX28*, and *AhBBX38* increased, while the expression level of *AhBBX25* increased and then decreased. Under the high salt treatment of the eight genes, except *AhBBX7*, *AhBBX15*, and *AhBBX28*, the expression levels of *AhBBX6*, *AhBBX11*, *AhBBX13*, *AhBBX25*, and *AhBBX38* increased. Together, all eight genes were found to be responsive to salt and/or drought treatments, with *AhBBX6*, *AhBBX11*, *AhBBX13*, and *AhBBX38* all showing increased expression under salt and drought stress, suggesting their potential involvement in regulating tolerance against salt and drought stresses.

### 2.10. Subcellular Localization Analysis of AhBBX6

The predicted subcellular localization results showed that *AhBBX6* is localized in the nucleus. To verify the location of *AhBBX6* in the cell, we performed subcellular localization experiments by transiently expressing the AhBBX6-GFP fusion protein and 35S-GFP (empty vector) in tobacco epidermal cells. The results showed that AhBBX6-GFP was expressed only in the nucleus, indicating that *AhBBX6* was localized in the nucleus (Figure 10).

### 2.11. Silencing of AhBBX6 Reduces Tolerance to Salt and Drought Stresses

We used virus-induced gene silencing to explore the potential function of *AhBBX6* in salt and drought tolerance. The octahydro lycopene desaturase (PDS) gene in peanuts (*Arahy.M5MKEZ*) was chosen as an indicator gene for the success of the VIGS system [31,32]. Based on qRT-PCR, the relative expression of PDS-encoding genes was significantly decreased in *AhPDS*-silenced plants compared to the control plants (*pTRV2/0*) (Appendix A).

The qRT-PCR results showed that the expression level of *AhBBX6* was reduced in *AhBBX6*-silenced plants (*pTRV2/AhBBX6*) compared with the control plants, suggesting the effective suppression of *AhBBX6* (Appendix A). Under the conditions of no salt or drought, the control and gene-silenced plants exhibited no significant disparity (Appendix A). However, after the treatment of salt stress, the leaves of *pTRV2/AhBBX6* plants curled and wilted, which were not observed in the control plants (*pTRV2/0*) (Figure 11A). We then determined the levels of antioxidant (CAT, SOD, and POD) and oxidant (MDA) enzyme concentrations in the control and *AhBBX6*-silenced plants. Compared to the control plants, the silenced plants had reduced concentrations of antioxidants and increased concentrations of MDA (Figure 11B–E).

After drought stress treatment, the leaves of *pTRV2/AhBBX6* plants showed significant crumpling and wilting compared to those of the control plants (Figure 12A), and the concentration of antioxidants (CAT, SOD, and POD) was significantly reduced in the silenced plants, whereas the concentration of MDA increased (Figure 12B–E). These results suggest that *AhBBX6*-silenced plants are more sensitive to salt and drought stresses.

## 3. Materials and Methods

### 3.1. Identification of AhBBX Members

The protein sequences of *A. hypogaea*, *A. duranensis*, and *A. ipaensis* were obtained from Peanutbase (https://legacy.peanutbase.org/peanut_genome (accessed on 14 March 2023)). The HMM profile for the zinc finger domain of the BBX protein (PF00643) [33] was acquired from the Pfam website and employed in the identification of *AhBBXs* using TBtools (version 2.069) [34]. The domains of BBX proteins were further examined using tools available on NCBI [35] and Pfam [36], and the location of the B-box 1/B-box 2/CCT structural domain was determined. A further manual examination was conducted to remove the candidate BBXs lacking the B-box domain to obtain the final set of peanut BBXs.

The online program ExPASy was used to determine the molecular weight (Mw) and isoelectric point (pI) of BBX proteins [37]. Based on Cell-PLoc 2.0, the subcellular localization of BBXs was predicted [38].

### 3.2. Sequence Alignment and Phylogenetic Analysis

BBX proteins from peanuts, *Arabidopsis* (https://www.arabidopsis.org/index.jsp (accessed on 14 March 2023)), and rice (https://riceome.hzau.edu.cn/ (accessed on 14 March 2023)) were aligned using the multiple sequence alignment function of Clustal W. The neighbor-joining (NJ) method of MEGA X software (version 10.2.6) was used to construct a phylogenetic tree with 1000 bootstrap replicates [39]. The tree was further edited using the R programming language.

### 3.3. Chromosomal Localization and Gene Duplication Analyses

The GFF file downloaded from PeanutBase was used to extract the chromosomal positions and lengths of *AhBBXs*, and the chromosomal positions of the genes were visualized using the online tool MG2C [40]. Gene duplication events and collinearity analysis were performed using TBtools software (version 2.069). The results of the gene duplication analysis and collinearity analysis were visualized using TBtools (version 2.069) [34].

### 3.4. Analysis of Gene Structure and Conserved Motif of BBX Proteins

The CDS and genomic sequences of peanut *BBXs* were obtained from PeanutBase. The TBtools software (version 2.069) was used to analyze gene structures. The conserved motifs of *BBXs* were analyzed using TBtools software (version 2.069) with parameters set to a maximum of 10 motifs, each of which was 6–50 amino acid residues in length. The conserved motifs were then mapped using TBtools software (version 2.069) [34].

### 3.5. Prediction of cis-Acting Elements in Promoter Sequences

To gain a better understanding of the *cis*-acting elements present in the promoter region of *AhBBXs*, the promoter sequences (2000 bp sequences located upstream of the transcription start site) of the genes were downloaded from PeanutBase and used for the analysis of *cis*-regulatory elements. The analysis was carried out utilizing the online program PlantCARE [41], and TBtools software (version 2.069) was used to visualize the findings [34].

### 3.6. RNA-Seq-Based Expression Profiling of AhBBXs

To ascertain the expression profile of peanut *BBXs* across different tissues and in response to abiotic stresses, we acquired publicly available peanut RNA-seq data from NCBI and PeanutBase and investigated the expression profile of *AhBBXs* in 22 different developmental stages and tissues [42], as well as under high salt and drought stresses [28,29]. The expression data of *AhBBXs* were filtered and normalized and then visualized by TBtools software (version 2.069) [34].

### 3.7. Abiotic Stress Treatments

The seeds of the cultivated peanut (*A. hypogaea* L.) variety Jinghua11 were provided by the Variety Resource Group of the Peanut Research Institute, Shandong, China. Seed germination was performed in a light incubator. When seedlings reached 3–4 fully expanded true leaves, they were treated with 200 mmol L^−1^ of NaCl [28] or 20% PEG (MW: 6000) [29]. The leaves and roots of the plants were collected at 0, 2, 6, 12, and 24 h post-treatment, with leaves and roots from the untreated plants serving as controls. Three biological replicate samples were collected for each time point. The collected specimens were promptly preserved in liquid nitrogen and kept at −80 °C for subsequent qRT-PCR analysis.

### 3.8. Total RNA Extraction, Reverse Transcription and qRT-PCR

Total RNA was extracted using the MiniBEST Plant RNA Extraction Kit (TaKaRa) and reversely transcribed to generate first-strand cDNA using the EScript RT-PCR (TaKaRa). The qRT-PCR was performed using the PerfectStart Green qPCR SuperMix kit. The reaction system and conditions were described by Mou et al., 2022 [43]. The qRT-PCR was carried out on a LightCycler 480 II instrument. Gene expression levels were calculated using the 2^−ΔΔC^ method [44]. Three biological replicates were used for each PCR assay. The peanut Actin gene [45] served as the internal reference gene.

### 3.9. Subcellular Localization of AhBBX6

The amplified *AhBBX6* coding sequence was first ligated into the pBWA(V)HS-GFP vector using the in-fusion method and transfected into *E. coli*-competent cells (5α) to construct the pBWA(V)HS-AhBBX6-GFP vector plasmid. The constructed vector plasmid generated was transferred into the *Agrobacterium tumefaciens* strain GV3101 and cultivated at 30 °C for 2 d. The suspension OD value was adjusted to 0.6, and then the cultured strain was injected into the lower epidermis of vigorously growing tobacco leaves. After cultivation, under low light conditions for 2 d, the injected leaf samples were taken to make slides and then observed under a laser confocal microscope (Nikon C2-ER, Tokyo, Japan). The GFP vector without *AhBBX6* was used as a control.

### 3.10. Virus-Induced Gene Silencing (VIGS) of AhBBX6

In order to study the function of *AhBBX6* under salt and drought tolerance, we used virus-inducted gene-silencing techniques to silence the gene. The peanut variety chosen was the cultivated peanut (*A. hypogaea* L.) variety Huayu9306 provided by the Variety Resource Group of the Peanut Research Institute, Shandong, China. The VIGS vectors, pTRV1 and pTRV2, were maintained by our laboratory. *pTRV2/AhBBX6* with restriction enzyme cleavage sites *Eco*RI and *Bam*HI was constructed. Plants were injected with the viral vector (*pTRV2/AhBBX6*) when they had grown to two fully expanded true leaves, and plants inoculated only with the TRV2 vector were utilized as controls. After injection, the plants were cultured in the dark for 3d and then placed in a light incubator. When the plants had their 5th fully expanded true leaf, the expression level of *AhBBX6* in the plants was analyzed using qRT-PCR. Plants with a gene silencing efficiency of more than 50% were regarded as true silencers and used in salt (200 mmol L^−1^ NaCl) and drought (20% PEG (MW: 6000)) treatments. Plants treated with water were used as controls.

### 3.11. Adversity Assay of Gene-Silenced Plants

Plants treated with slat or drought stress were subjected to the detection of MDA (malondialdehyde) content, SOD (catalase), POD (peroxidase), and CAT (catalase) activities using commercial assay kits (Geruisi Bio, Suzhou, China). For each assay, three biological replications were conducted.

## 4. Discussion

The *BBX* gene family is a class of zinc-finger transcription factors that are widely distributed in plants and exert significant influence on plant development and responses to various stresses. The identification of *BBX* family members has been documented in multiple plant species, highlighting the significance of the gene family [1,2,3,4,5,6,7,8].

On the basis of the identification of 77 *BBXs* in three peanut species, this study performed a comprehensive phylogenetic analysis of *BBXs* in three peanut species, *Arabidopsis* and rice. The analysis classified the 77 peanut BBXs into five subfamilies (Figure 1, Appendix A), which is consistent with previous research conducted in *Arabidopsis* [1]. Members within the same subfamily exhibit conserved or similar distributions of protein motifs (Figure 5B, Appendix A), supporting the results of the evolutionary analysis and also suggesting that peanut *BBX* members within the same subfamily might have similar biological functions. Previous studies have shown that the intron–exon structure can provide support for the phylogenetic relationships of gene families [46]. In this study, genes on the same sub-branch of the phylogenetic tree generally possessed comparable exon–intron structures (Figure 5C, Appendix A). Furthermore, gene duplication events could potentially play a role in the expansion of gene families [26]. In this study, 43 segmental duplicated gene pairs were identified among *AhBBXs* (Figure 3, Appendix A), indicating that each duplicated gene pair may have relatively close evolutionary relationships and gene functions.

Salt stress and drought stress can severely affect crop growth and yield [47,48,49]. Currently, it has been reported that a variety of plants have been able to achieve high or stable yields under drought and high salt conditions by increasing or decreasing the expression of some of their own genes to withstand adverse external conditions. For instance, *AhABI4*-silenced plants exhibited increased survival rates under salt stress [32]. Overexpressing the *TaGPX1-D* [50] or *TaNCL2-A* [51] of wheat (*Triticum aestivum*) in *Arabidopsis* enhanced *Arabidopsis* tolerance to salt and osmotic stresses. In addition, the overexpression of the zinc finger protein *GmZF351* has been reported to improve the tolerance of soybeans to salt and drought stress [52]. Numerous studies have demonstrated that *BBXs* have a vital function in controlling the complex mechanisms of abiotic stress response [30,53].

The analysis of transcriptome data from previous studies revealed that 11 *AhBBXs* had a two-fold increase in expression under drought stress, while six *AhBBXs* exhibited a two-fold increase in expression under salt stress (Figure 8, Appendix A). The stress responses of some of these genes were validated by qRT-PCR (Figure 9). Previous studies have found that *AtBBX24* is involved in salt stress signal transduction [54]. Plants overexpressing *AtBBX24* show improved salt tolerance compared to wild-type *Arabidopsis* [55], and *AhBBX13* belongs to the same cluster as *AtBBX24*, which is highly expressed under salt stress. Therefore, it is hypothesized that *AhBBX13* is likely to be related to responses to salt stress. The complex of *CmBBX19* and ABF3 (ABA signaling) is capable of regulating plant drought tolerance via the ABA-dependent pathway [18]. *OsBBX2* and *OsBBX24* of subfamily V are involved in regulating signal transduction pathways under cold and drought stresses [19]. *AhBBX15*, located in the same subfamily as these two rice genes, is highly expressed under drought stress. Thus, it is speculated that *AhBBX15* might play a role in drought tolerance in peanuts. The overexpression of *AtBBX5* leads to increased resistance to salt stress in genetically modified *Arabidopsis* through the abscisic acid-dependent signaling pathway [56]. In this study, *AhBBX38* clustered together with *AtBBX5* and showed a differential expression under salt stress, suggesting the potential functional similarity between *AhBBX38* and *AtBBX5* and the potential involvement of *AhBBX38* in the salt stress response.

We discovered significant increases in *AhBBX6* expression after both salt and drought treatments (Figure 9) and *AhBBX6*-silenced plants were more sensitive to salt and drought treatments (Figure 11 and Figure 12), indicating the role of this gene in salt and drought tolerance. Previous studies have demonstrated that the activities of the antioxidant enzyme system are positively correlated with plant salt and drought tolerance [57]. These enzymes not only protect various components of the cell from damage but also play a vital role in plant growth and development by regulating cell–subcellular processes [58,59,60,61]. Under salt or drought stress, the antioxidant contents decreased significantly in *pTRV2/AhBBX6* plants, and the MDA content increased significantly, suggesting that the integrality of the plasma membrane is associated with *AhBBX6*-meditated salt and drought tolerance.

Overall, *AhBBXs* likely play crucial roles in controlling how peanuts react to various stresses, and *AhBBX6* positively regulates peanut tolerance to salt and drought stresses. Further studies are required to elucidate the precise functions and fundamental mechanisms of *AhBBXs* in peanut growth and stress responses. The findings of this study establish a fundamental basis for selecting candidate *AhBBXs* for further investigation.

## 5. Conclusions

In this study, 77 *BBXs* were identified from wild and cultivated peanut species. The newly identified *BBX* family members were comprehensively analyzed using bioinformatics methods such as phylogenetic analysis, protein-conserved motifs, chromosomal localization, intraspecific collinearity, and expression patterns. Furthermore, four *BBXs* were identified as candidate regulators of tolerance to saline and drought stresses. Notably, the silencing of *AhBBX6* reduced tolerance to salt and drought stress in peanuts. These findings contribute information for further studying the function and regulatory mechanisms of *BBXs* in peanuts under salt or drought stress.

## Figures and Tables

**Figure 1 plants-13-00955-f001:**
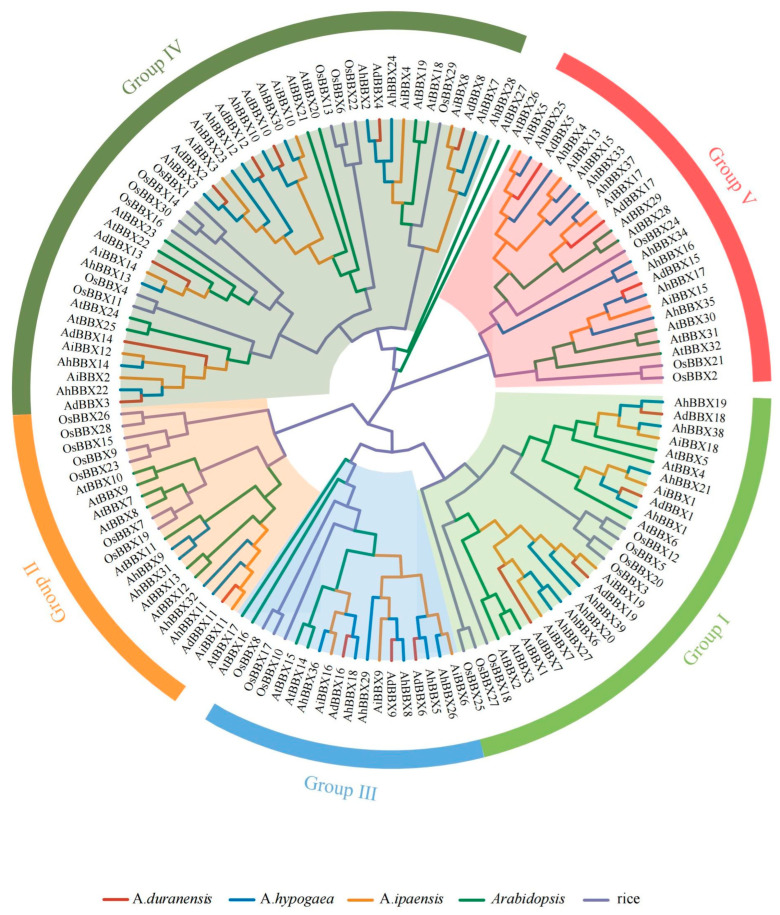
The phylogenic tree of BBX proteins from different species. The BBXs were classified into five subfamilies indicated by different colors on the outermost circle. The BBXs from different species were color-coded with red lines representing *A. duranensis*, blue lines representing *A. hypogaea*, orange lines representing *A. ipaensis*, green lines representing *Arabidopsis*, and purple lines representing rice. *At*: *Arabidopsis*, *Os*: rice, *Ad*: *A. duranensis*, *Ai*: *A. ipaensis*, and *Ah*: *A. hypogaea*.

**Figure 2 plants-13-00955-f002:**
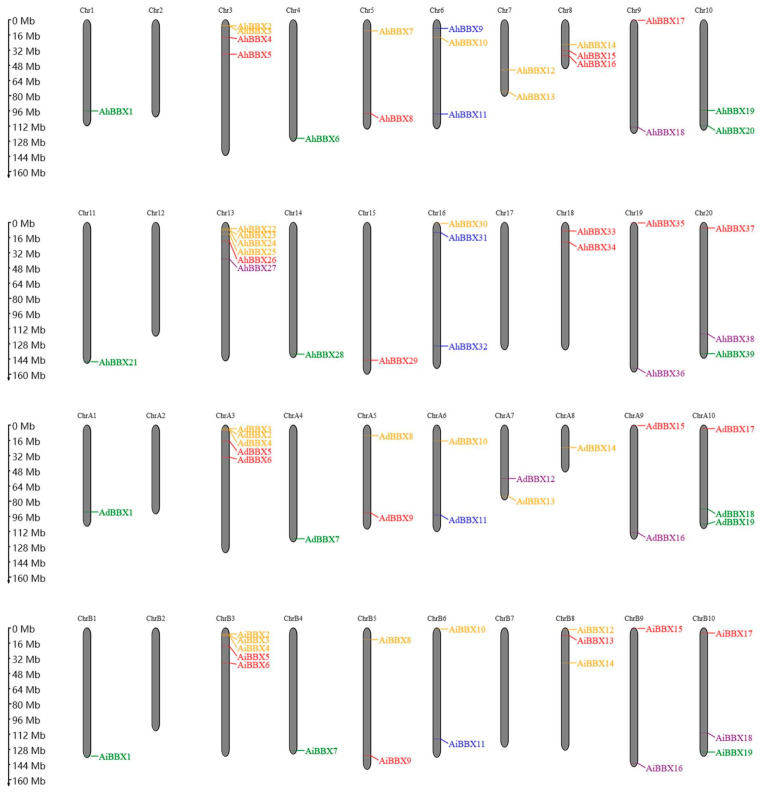
The chromosomal location of *BBXs*. The positions of the 77 peanut *BBXs* are displayed on the right side of the chromosomes. *BBXs* of different subfamilies are indicated by different colors. The top two panels show the chromosomes of the tetraploid cultivated peanut species, and the bottom two panels show the chromosomes of the two wild peanut species.

**Figure 3 plants-13-00955-f003:**
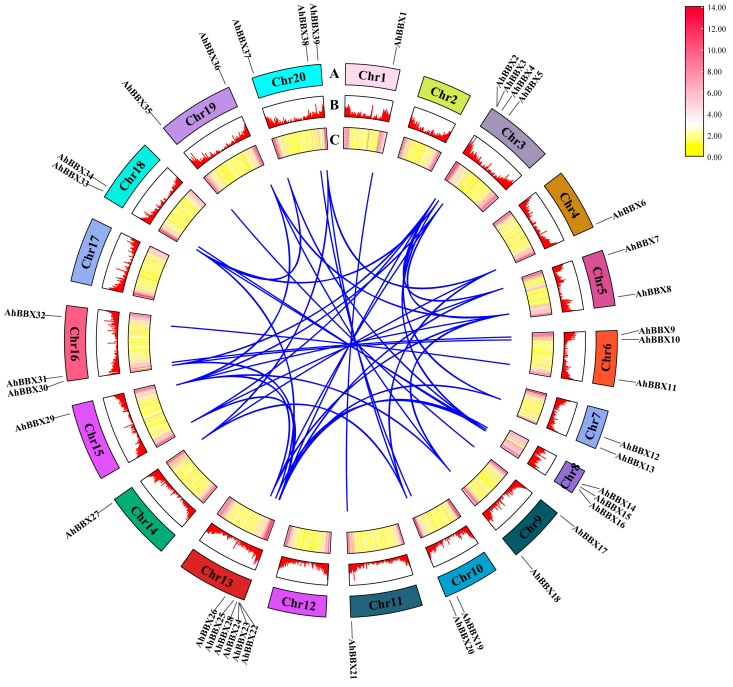
Segmental duplication analyses of *AhBBXs*. Blue lines indicate the potential duplication relationship between *AhBBXs*. (A) Positions of *AhBBXs* on chromosomes (AhChr1-AhChr20) with different chromosomes are indicated by different colors. The chromosomes were scaled based on their lengths. (B) Bar graphs showing gene density on each chromosome, with higher bars representing higher gene density. (C) Heat map showing the distribution of gene density on each chromosome, with red and yellow colors representing high and low densities, respectively.

**Figure 4 plants-13-00955-f004:**
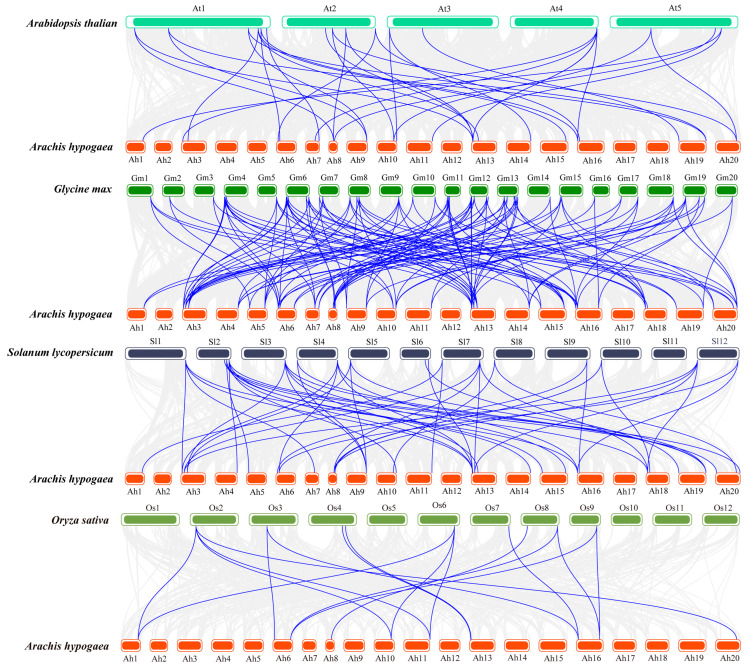
The syntenic analysis of *AhBBXs* among the four species. The grey lines represent the collinear blocks among the peanuts and various species. The blue lines represent the collinearity between *AhBBXs* and various species.

**Figure 5 plants-13-00955-f005:**
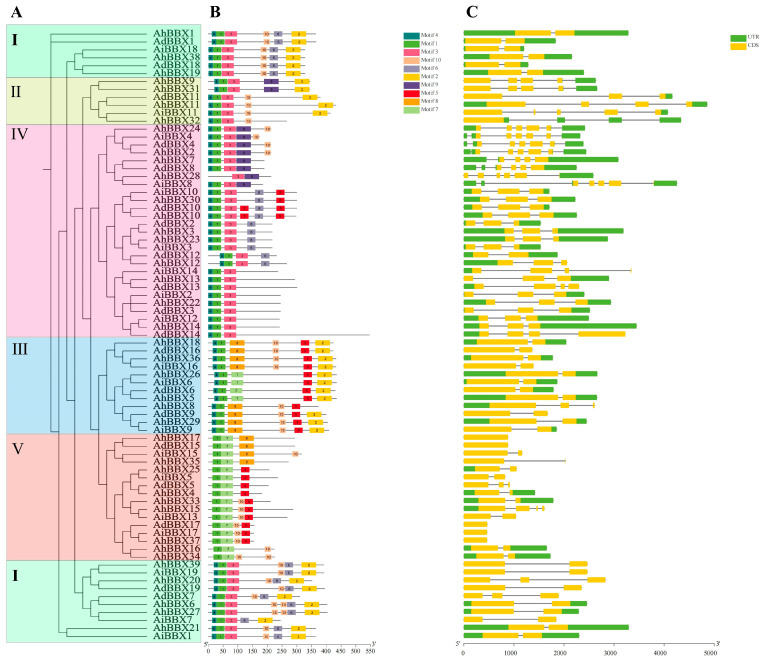
Gene structure and conserved motifs of peanut BBXs from different subfamilies (I–V). (**A**) The phylogenetic tree constructed with MEGA X, categorizing BBXs into five subfamilies denoted by Roman numerals (I–V). Different subgroups are represented by differently colored boxes. (**B**) The conserved motifs in each of the BBX proteins, with ten motifs indicated by distinctly colored boxes. (**C**) Gene structures of each *BBX* gene. Yellow and green boxes represent the exons and UTR, respectively, and grey lines depict introns.

**Figure 6 plants-13-00955-f006:**
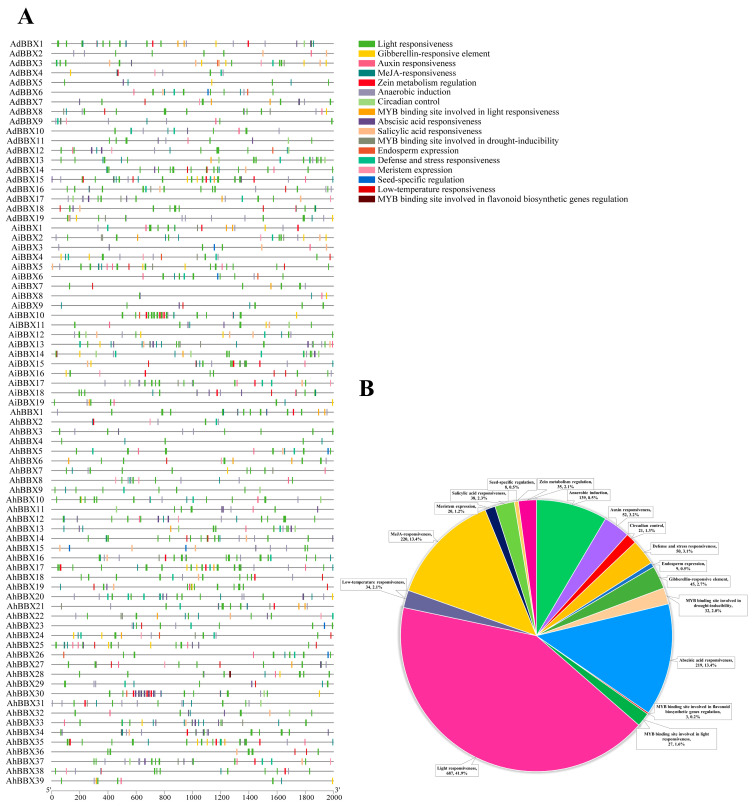
The *cis*-acting regulatory elements within the promoters of *BBXs*. (**A**) Distribution of *cis*-acting elements in the promoters of peanut *BBXs*. (**B**) A pie chart visually depicts the proportions of various *cis*-acting elements in peanut *BBXs*.

**Figure 7 plants-13-00955-f007:**
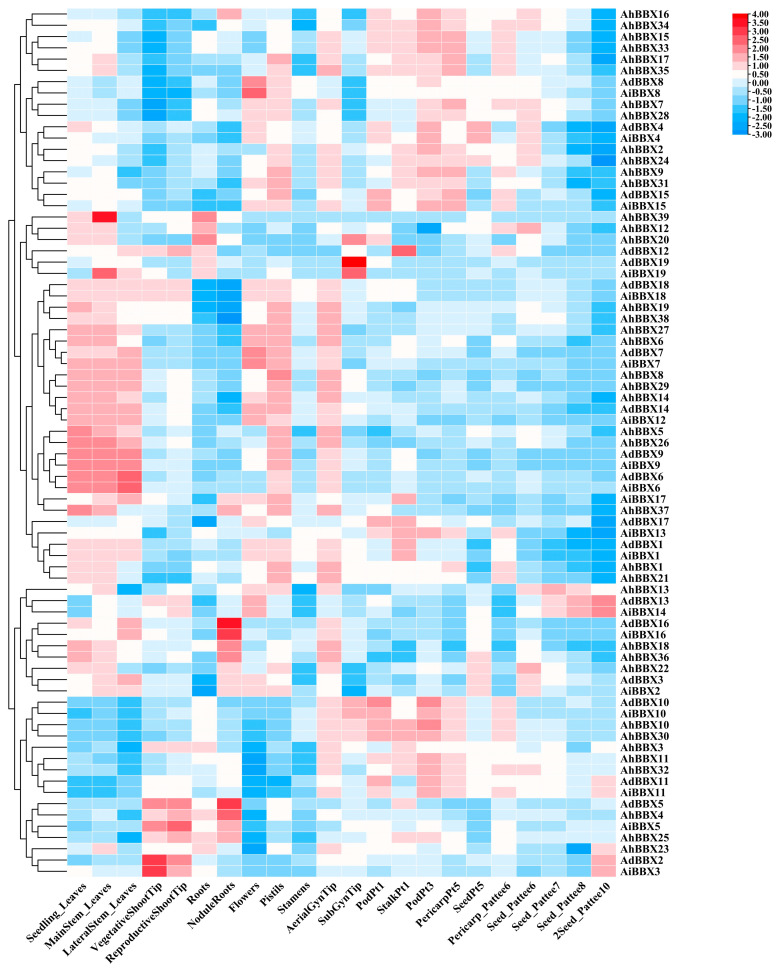
Expression profiles of *BBXs* in 22 tissues of peanuts. The heatmap was generated using TBtools software, with the log2−transformed FPKM values of *BBXs*. The red and blue colors represent high and low expression values, respectively.

**Figure 8 plants-13-00955-f008:**
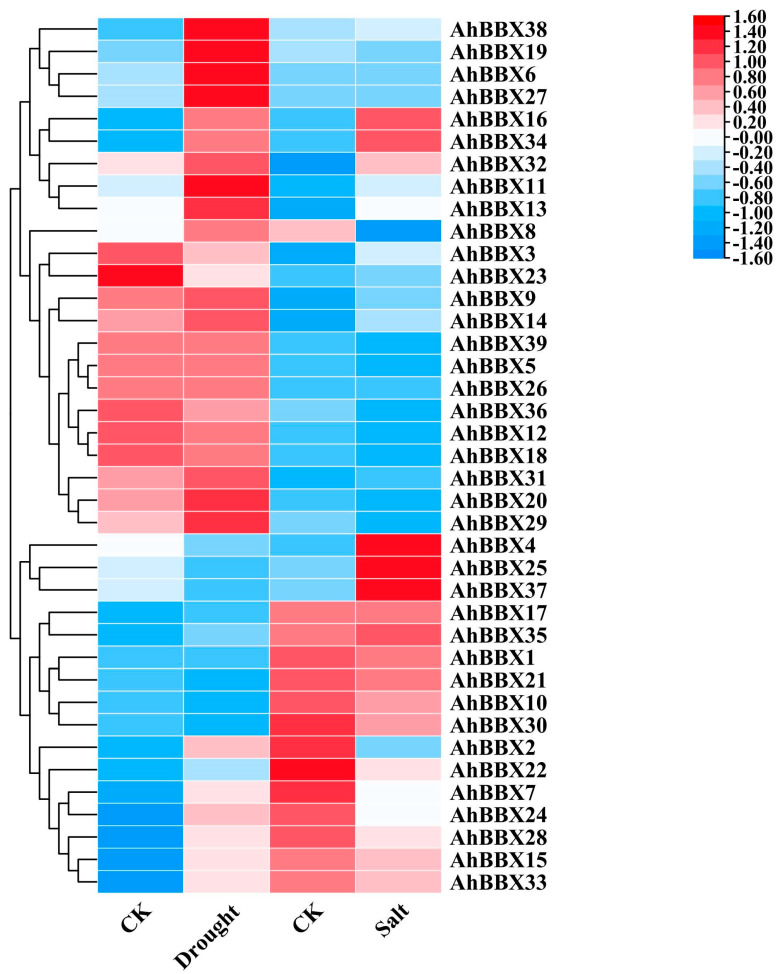
The changes in the expression levels of *AhBBXs* in response to salt or drought stress treatments based on the RNA−seq data reported in references [28,29]. CK (control) indicates the expression level of *AhBBXs* without stress treatment. Drought indicates the expression level of *AhBBXs* in the mixed plant samples collected at 6 h, 12 h, 24 h and 48 h after treatment with 20% PEG (MW: 6000). Salt indicates the expression level of *AhBBXs* in plant samples at 24 h after treatment with 200 mmol L^−1^ NaCl. The FPKM values of *AhBBXs* were log2-transformed and used to generate a heatmap with TBtools software. The red and blue colors represent the higher and lower relative abundance of *AhBBXs*, respectively.

**Figure 9 plants-13-00955-f009:**
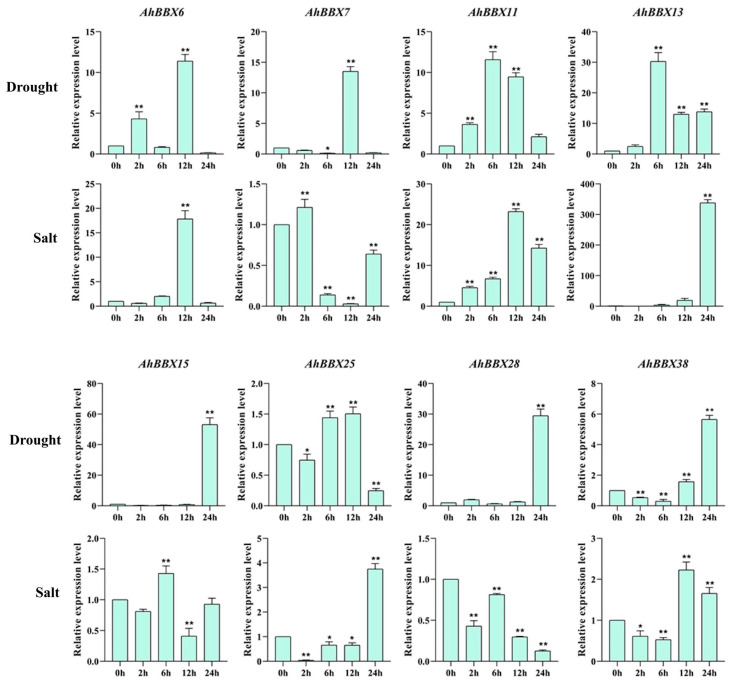
Comparison of the expression of 8 *AhBBXs* uner drought or salt stress based on qRT-PCR. The data are presented as the mean ± SD (n = 3). The * and ** symbols represent a significant difference compared to 0 h at *p* < 0.05 and *p* < 0.01, respectively.

**Figure 10 plants-13-00955-f010:**
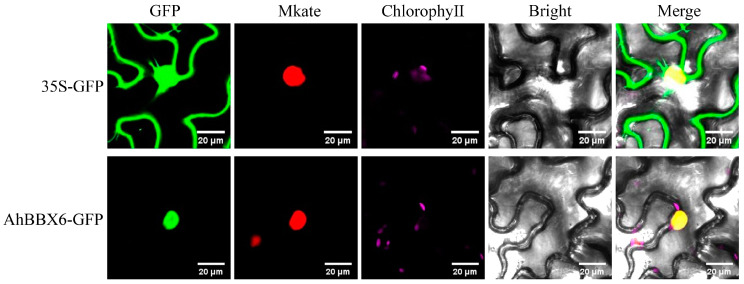
Subcellular localization of *AhBBX6* in tobacco leaves. The green fluorescence in the GFP plot represents the GFP fluorescence signal. The red fluorescence in the Mkate plot represents the nucleu marker. The purple fluorescence in the ChlorophyeⅡ plot represents the Chloroplast fluorescence signal. The Bright plot is the bright field. The yellow fluorescence in the Merge plot represents the overlap of the green fluorescence from GFP with the red fluorescence from the marker. The visible light, and merged green fluorescence and visible light images are presented.

**Figure 11 plants-13-00955-f011:**
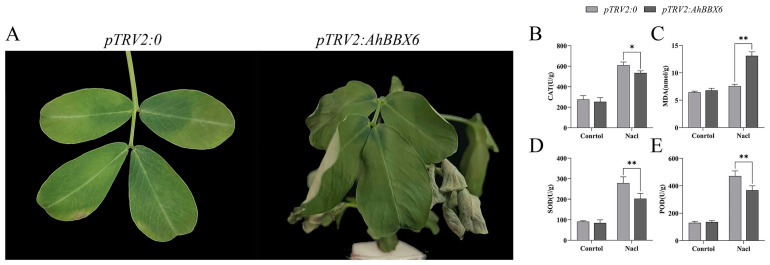
Silencing of *AhBBX6* reduces tolerance to salt in peanuts. (**A**) Phenotype of the control (*pTRV2/0*) and gene-silenced (*pTRV2/AhBBX6*) plant after salt treatment. (**B**–**E**) Comparison of CAT (**B**), MDA (**C**), SOD (**D**), and POD (**E**) activities before and after salt treatment. The * and ** symbols indicate significant differences at *p* < 0.05 and *p* < 0.01, respectively.

**Figure 12 plants-13-00955-f012:**
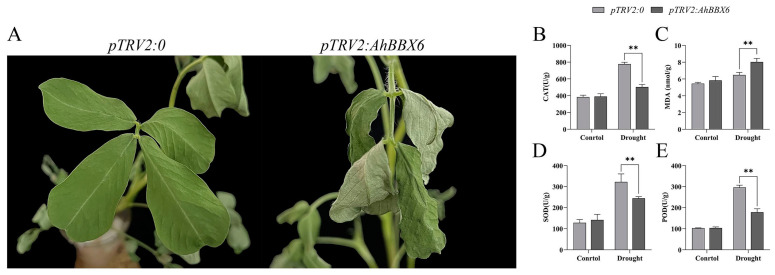
Silencing of *AhBBX6* reduces tolerance to drought in peanuts. (**A**) Phenotype of the control (*pTRV2/0*) and gene-silenced (*pTRV2/AhBBX6*) plant after drought treatment. (**B**–**E**) Comparison of CAT (**B**), MDA (**C**), SOD (**D**) and POD (**E**) activities before and after drought treatment. The ** symbol indicates significant differences at *p* < 0.01, respectively.

## Data Availability

All data supporting the findings of this study are available within the paper and within its Appendix A published online.

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
