# Peer review of "Genome-Wide Identification of Peanut B-Boxs and Functional Characterization of AhBBX6 in Salt and Drought Stresses"

_plants, 2024, doi:10.3390/plants13070955_

Round 1
Reviewer 1 Report
Comments and Suggestions for Authors
The studies of the identification of peanut BBXs and the functional characterization of the AhBBX6 under salt and drought condition is well presented by the authors. There are relevant analysis that shows how BBXs transcription factors are involved in the development of peanut and also are involved in stress conditions.
The manuscript is well written, however, I consider that some description should be better presented:
Line 137: 20% PEG was used as drought condition and 200 mmol/L of NaCl were used for the stress condition. A previous experiment was done to show that those amount are the best for the stress studies in peanut? if it was done, explain or add a reference.
Line 150-173: the space between line are different from the rest of the manuscript. Please format.
Line 166-176: all the RT-PCR and enzyme analyses were done with samples isolated at the same day? how many days after virus vector inoculation to obtain the 5th true leaf was fully expanded? How many plants were analyzed? The results are the mean of how many plants?
Lines 332-334: the legend of figure 7 should describe the experiment. What is the meaning of CK? What is the amount of salt? What is the drought condition?
Also I consider that all the figure legends should be reviewed describing what the figure represents and what the authors would like to be shown.
Author Response
Thanks very much for your comments

Reviewer 2 Report
Comments and Suggestions for Authors
The article entitled, “Genome-wide Identification of Peanut BBXs and Functional Characterization of AhBBX6 in Salt and Drought Stresses” submitted by Dr. Jie Sun, needs major revision and improvement in the scientific writing and language of the manuscript.
- There are many grammatical errors throughout the manuscript.
- A thorough reading is required.
- The abstract part is not crisp and does not connect well, and there is more scope for language improvement in this section.
- The introduction part is also long and general.
- The Material and method part requires scientific writing
- The Result part also needs many improvements. The authors claim to perform VIGS. However, no PDS control has been used. It raises concerns regarding functional characterization. How do authors defend this?
- The Discussion part is shallow and repeats of result portion. The addition of more citations and their concepts is needed in the discussion portion.
- What are other genes reported and functionally demonstrated for drought and salt tolerance should be discussed, for instance TaGPX1-D and TaNCL2-A genes expression is also shown for such tolerance in transgenic Arabidopsis. Similarly, DPY1 is known as an osmosensor for drought signaling. These should be the part of discussion.
Moderate editing of English language required
Author Response
Thanks very much for your comments.

Reviewer 3 Report
Comments and Suggestions for Authors
The authors conducted a comprehensive bioinformaic study on 77 B-box (BBX) genes from wild and cultivated peanut species A.duranensis, A.ipaensis, A.hypogea that they determined using genome mining. They conducted Phylogenetics analysis on the three peanut species, Arabidopsis, and rice BBXs to classify the proteins into different groups/sub families. Further they conducted bioinformatic analysis to identify the chromosome distribution, exon-intron structures, function cis-regulatory elements involved in plant development, light, and hormone response. Additionally, they conducted gene expression analysis of peanut BBXs and discovered crucial peanut BBX genes that are important for development, hormone response and abiotic stresses (salt and drought). They also conducted qRT-qPCR analysis on eight peanut BBXs and showed the expression of these BBXs change in response to drought or high salt stress, indicating their importance under salt stress.
Furthermore, authors did gene silencing experiment for the functional validation of AhBBX6 that showed a high expression during salt and drought stress. They determined that silencing of AhBBX diminishes the resistance of the peanut plants to salt and drought stress.
Overall, it is a comprehensive bioinformatics study and understanding of peanut BBXs. This information is crucial for further functional validation of peanut BBXs during abiotic stresses and plant growth. The manuscript is well-written and comprehensive.
Comments on the Quality of English LanguageThe manuscript is well-written and easy to follow. However, authors are encouraged to proof read the manuscript for any minor grammatical errors.
Author Response
Thanks very much for your comments.

Round 2
Reviewer 2 Report
Comments and Suggestions for Authors
The Ms may now be accepted